# Degradable mesoporous semimetal antimony nanospheres for near-infrared II multimodal theranostics

Yu Chen[1,6], Zhongzheng Yu [2,6], Kai Zheng[3], Yaguang Ren[4], Meng Wang[1], Qiang Wu[1], Feifan Zhou[1], Chengbo Liu [3], Liwei Liu [1], Jun Song [1✉] & Junle Qu [1,5✉]

Metallic and semimetallic mesoporous frameworks are of great importance owing to their unique properties and broad applications. However, semimetallic mesoporous structures cannot be obtained by the traditional template-mediated strategies due to the inevitable hydrolytic reaction of semimetal compounds. Therefore, it is yet challenging to fabricate mesoporous semimetal nanostructures, not even mention controlling their pore sizes. Here we develop a facile and robust selective etching route to synthesize monodispersed mesoporous antimony nanospheres (MSbNSs). The pore sizes of MSbNSs are tunable by carefully controlling the partial oxidation of Sb nuclei and the selective etching of the as-formed $Sb_2O_3$. MSbNSs show a wide absorption from visible to second near-infrared (NIR-II) region. Moreover, PEGylated MSbNSs are degradable and the degradation mechanism is further explained. The NIR-II photothermal performance of MSbNSs is promising with a high photothermal conversion efficiency of ~44% and intensive NIR-II photoacoustic signal. MSbNSs show potential as multifunctional nanomedicines for NIR-II photoacoustic imaging guided synergistic photothermal/chemo therapy in vivo. Our selective etching process would contribute to the development of various semimetallic mesoporous structures and efficient multimodal nanoplatforms for theranostics.

[1] Center for Biomedical Optics and Photonics (CBOP) & College of Physics and Optoelectronic Engineering, Key Laboratory of Optoelectronic Devices and Systems of Guangdong Province and Ministry of Education, Shenzhen University, Shenzhen 518060, China. [2] School of Chemical and Biomedical Engineering Nanyang Technological University, 637459 Singapore, Singapore. [3] Northwestern Polytechnical University, School of Civil Aviation, 127 West Youyi Road, Beilin District, Xi'an, Shanxi 710072, China. [4] Research Laboratory for Biomedical Optics and Molecular Imaging, Shenzhen Institutes of Advanced Technology, CAS Key Laboratory of Health Informatics, Chinese Academy of Sciences, Shenzhen 518055, China. [5] National Research Nuclear University MEPhI (Moscow Engineering Physics Institude), Moscow 115409, Russian Federation. [6]These authors contributed equally: Yu Chen, Zhongzheng Yu. ✉email: songjun@szu.edu.cn; jlqu@szu.edu.cn

Mesoporous nanomaterials with large surface areas, tunable pore sizes, and diverse compositions have drawn tremendous interest due to their great potential in various applications, including drug delivery, catalysis, energy conversion and storage, as well as separation technique[1–6]. In the past two decades, numerous silica-based mesoporous structures with tunable pore sizes and morphologies were constructed by a modified stöber process[7–13], but the inert properties of silica seriously limited their applications. Therefore, metallic and semimetallic mesoporous frameworks are expected to show superior performance than silica-based mesoporous structures in electronics, catalysis, optics, and biophotonics, owing to the extraordinary electronic properties of metals and semimetals[14–17]. Several metallic mesoporous nanostructures, including platinum, palladium, gold and rhodium etc. have been successfully prepared in recent years[18–22]. The synthesis methods for metallic mesoporous nanostructures were currently limited to templated-mediated method or wet chemical method[23]. But the formation mechanism and controllable synthesis of these metallic mesoporous structures accompanying with demonstrations in various applications have attracted huge attention[24–26]. In contrast, semimetallic mesoporous structures have rarely been reported[27]. We attribute the main obstacle to the inevitable hydrolytic reaction and rapid precipitation of semimetal compounds (arsenic, antimony and bismuth) that result in the lack of necessary interaction between the semimetal precursor and pore-forming template. Hence, a facile and robust approach for the controllable synthesis of semimetallic mesoporous nanomaterials is still in urgent need and of great challenge.

Furthermore, semimetals are reported to show decent photothermal performance in the second near-infrared (NIR-II, 1000–1700 nm) range[28–31]. Semimetal-based photothermal agents (PTAs) can absorb the energy of NIR-II light and convert it to heat for photothermal therapy (PTT), and simultaneously generate photoacoustic signals for deep-tissue photoacoustic imaging (PAI)[32–36]. Semimetallic mesoporous nanomaterials are excellent candidates for multimodal theranostic agents with loading capacity and photothermal conversion ability. Recently, antimony (Sb) nanomaterials have been demonstrated to have a wide absorption range from visible to NIR-II region, remarkable photothermal conversion efficiency (PTCE) and intensive photoacoustic signal in the first near-infrared (NIR-I, 700–1000 nm) region[37–40]. There are few studies of Sb-based PTAs in the NIR-II range. Furthermore, Sb nanomaterials have shown the photothermal induced degradable property, which facilitated the fast clearance of Sb nanomaterials to avoid the long-term toxicity issue of nanomedicines. The structure design of Sb-based nanomaterials for biomedical applications has been focused on two dimensional (2D) materials, including 2D Sb quantum dots and especially antimonene[39–41], which are impossible to involve mesoporous structure. 3D Sb-based nanostructures, including Sb nanorods and nanopolyhedrons, have been demonstrated as NIR-I PTAs without mesopores[28,37]. Therefore, mesoporous Sb-based nanoconstructs with tunable pore sizes are expected to achieve NIR-II PAI guided synergistic therapy with safer and better therapeutic outcomes.

In this work, we describe a simple and effective route to prepare the monodispersed mesoporous Sb nanospheres (MSbNSs) with tunable pore sizes via a selective etching process. We systemically study the formation mechanism of MSbNSs and realize the controllable tuning of MSbNSs pore sizes. The resultant MSbNSs possess a high NIR-II PTCE up to 44%, intensive NIR-II photoacoustic signal, and photothermal-induced degradation. The photothermal degradation mechanism of MSbNSs is proposed, simulated and analyzed. Based on these findings, MSbNSs-based multifunctional nanomedicines are successfully constructed and presented for NIR-II PAI guided synergistic photothermal/chemo therapy in vivo. The PEGylated MSbNSs loaded with doxorubicin (DOX) show drug loading and release ability and NIR-II photothermal performance with elimination of tumors in mice. Our approach of selective etching would contribute to the development of more types of semimetallic mesoporous nanostructures and more multimodal theranostics by loading different drugs, antigens or mRNAs for multifunctional chemotherapy, immunotherapy or gene therapy with NIR-II guided PAI and PTT.

## Results

**Preparation, characterization and structural control of MSbNSs.** The schematic illustration of the formation process of MSbNSs is shown in Fig. 1a. Dodecylthiol (DDT) and oleylamine (OLA) were used as capping ligands to control the shape of Sb nanospheres, while octadecene (ODE) was used as solvent, and $SbCl_3$ and tert-butylamine borane were selected as semimetal precursor and reducing agent, respectively. After chemical reduction, numerous small Sb nuclei were formed, which were sensitive to oxygen ($O_2$) and could be easily oxidized due to their high surface energy. As the Sb nanocrystals grew, the oxidation product antimony oxide ($Sb_2O_3$) was dispersed in Sb nanocrystals, which could further react with DDT in the solution as the selective etching process of $Sb_2O_3$ to form mesopores. The pore sizes of MSbNSs are tunable by carefully controlling the oxidative degree of Sb nuclei and the selective etching of the as-formed $Sb_2O_3$. Detailed experimental procedures were provided in experimental section. The resultant MSbNSs have shown uniform spherical morphologies in the transmission electron microscopy (TEM, Supplementary Fig. 1), and their high-angle annular dark-field scanning TEM (HADDF STEM) images (denoted as MSbNSs-1, MSbNSs-2, MSbNSs-3 and MSbNSs-4) were shown in Fig. 1b–e. The sizes of MSbNSs were gradually increased from 45 nm, to 48 nm, 51 nm, and to final 63 nm (Supplementary Fig. 2), respectively. $N_2$ absorption and pore size analysis were performed to demonstrate the mesoporous structures of MSbNSs (Fig. 1f). The obvious hysteresis loop of the absorption/desorption isotherms in all of the cases implied the typical mesoporous structures of the resultant MSbNSs. The Brunauer-Emmett-Teller (BET) surface area of MSbNSs increased from $78 \, m^2 \, g^{-1}$ to $252 \, m^2 \, g^{-1}$, $233 \, m^2 \, g^{-1}$ and to final $216 \, m^2 \, g^{-1}$. The surface areas of obtained MSbNSs were smaller than the silica or carbon-based mesoporous structures, indicating a relatively lower porosity[42,43]. MSbNSs with different pore sizes were obtained via selective etching of SbNSs by controlling the amount of $O_2$ in the reaction precursor. By controlling the pumping time of $O_2$ from 10 s to 30 s, the average mesopore size of MSbNSs has been increased from 2.6 to 3.7 nm. Increase of $O_2$ amount has generated larger pores (~11.5 nm) in MSbNSs-3 besides small mesopores with average sizes of 3.9 nm and 5.7 nm. The complete curves of pore size distribution of MSbNSs were provided as shown in Supplementary Fig. 3. The nitrogen sorption isotherms of the MSbNSs-3 showed two major capillary condensation steps in the relative pressure ranges 0.1–0.3 and 0.75–0.98, respectively, indicating that at least two sets of pores coexisted in the nanospheres. MSbNSs-1, MSbNSs-2 and MSbNSs-3 can still remain integrated spherical shape. Nevertheless, further increased $O_2$ amount would destroy the nanostructure of MSbNSs and lead to collapsed MSbNSs-4 with larger pores (~18 nm). To further confirm the important role of $O_2$ in the formation of mesopores, a control experiment without the pumping of $O_2$ was performed under the situation that other experimental parameters were kept constant. Mesoporous structures were not observed as shown in Supplementary Fig. 4. MSbNSs with tunable mesopore sizes were successfully fabricated by our selective etching method. The shape of MSbNSs can

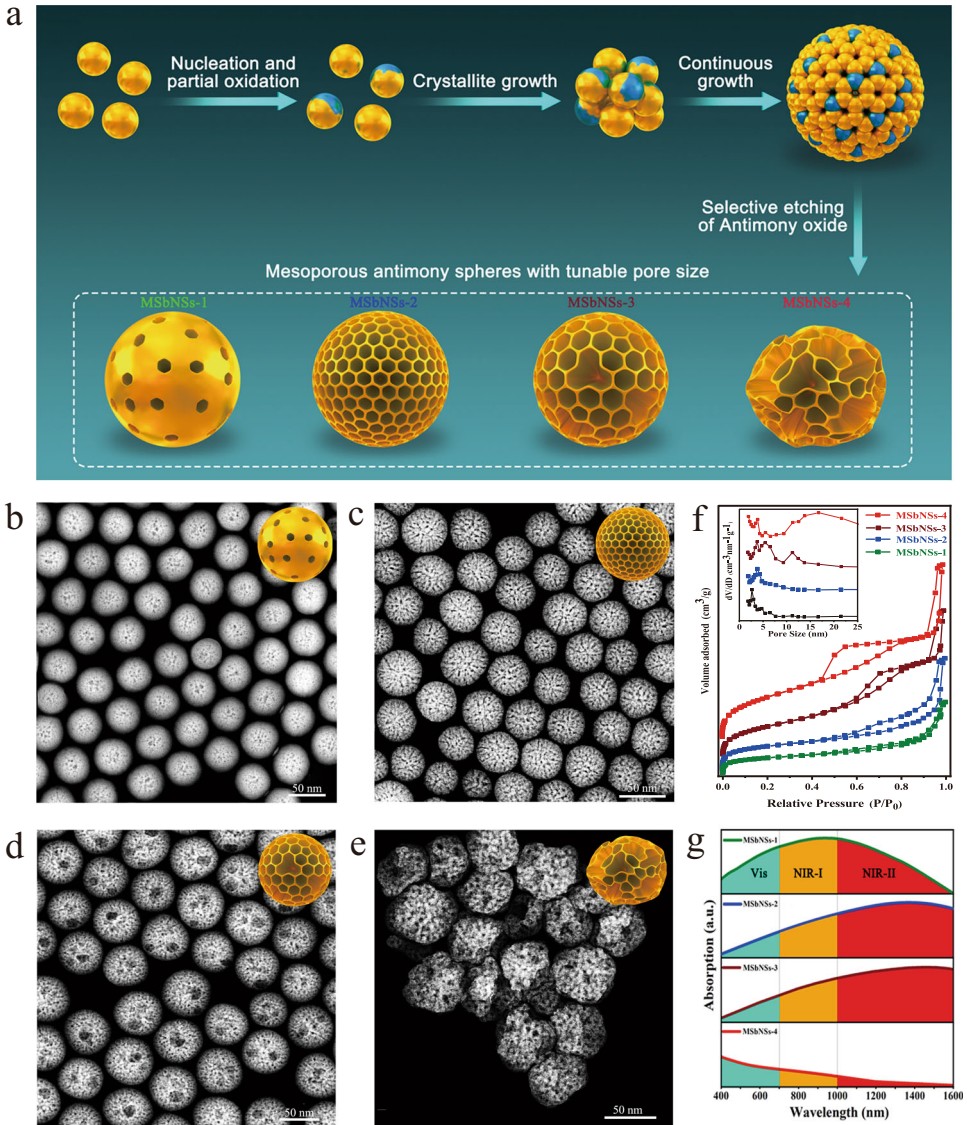

**Fig. 1 Fabrication and characterization of MSbNSs. a** Schematic illustration of the fabrication of MSbNSs with tunable pore sizes. STEM images of **b** MSbNSs-1, **c** MSbNSs-2, **d** MSbNSs-3, **e** MSbNSs-4. A representative image of 3 replicates from each group is shown in **b**–**e**. **f** $N_2$ physisorption isotherms and their corresponding pore sizes calculated by the BJH method. Experiments were performed two times with similar results. **g** Vis-NIR absorption spectra of MSbNSs-1/2/3/4. Experiments were performed three times with similar results.

also be facilely tuned by regulating the reaction temperature (Supplementary Fig. 5), in which the shape of the obtained products changed from sphere to polyhedron at higher reaction temperatures.

Moreover, MSbNSs-1, MSbNSs-2 and MSbNSs-3 showed strong and broad absorption, especially in the NIR-II range (Fig. 1g, Supplementary Fig. 6). From MSbNSs-1 to MSbNSs-3, the absorption peaks were further red-shifted towards NIR-II range, providing the potential for excellent NIR-II photothermal performance. The collapsed MSbNSs-4 lost the strong NIR-II absorption, which is not applicable for the multimodal theranostics.

**Formation mechanism of MSbNSs.** To shed light on the selective etching mechanism, X-ray diffraction (XRD) and X-ray photoelectron spectroscopy (XPS) were carried out to analyze the reaction process. XRD patterns revealed that MSbNSs-1 and MSbNSs-2 remained pure rhombohedral phase of Sb corresponding with the standard card JCPDS 085-1324, while a mixed phase of rhombohedral phase of Sb and cubic phase of $Sb_2O_3$ corresponding with JCPDS 05-0534 was detected in MSbNSs-3 and MSbNSs-4 (Fig. 2a). *XPS* results further illustrated that lower amount of $O_2$ first oxidized Sb to form $Sb_2O_3$ as measured in MSbNSs-1, while higher amount of $O_2$ would further generate $Sb_2S_3$ that derived from the reacting $Sb_2O_3$ with DDT, since $Sb_2S_3$ first appeared in MSbNSs-2 and gradually increased in MSbNSs-3 and MSbNSs-4 (Fig. 2b). $Sb_2O_3$ and $Sb_2S_3$ generated in the early oxidation process were not crystalized, thus not detectable in XRD patterns. $Sb_2O_3$ could crystalize with a cubic phase as the amount of $O_2$ increased. $Sb_2S_3$ will remain amorphous since the crystallization temperature of $Sb_2S_3$ is at least above 240 °C and our reaction temperature was 170 °C[44,45], thus not detectable by XRD. Moreover, $Sb_2S_3$ could react with $O_2$ to form $Sb_2O_3$ and $SO_2$, leading to a dynamic reaction cycle to accelerate the etching process. A higher amount of $O_2$ could even further oxidize $Sb_2O_3$ to $Sb_2O_5$ as $Sb3d_{5/2}$ only appeared in MSbNSs-4. To deepen the understanding the role of DDT, we studied the impact of DDT concentration on the morphology of MSbNSs. The concentration of DDT was varied while the other

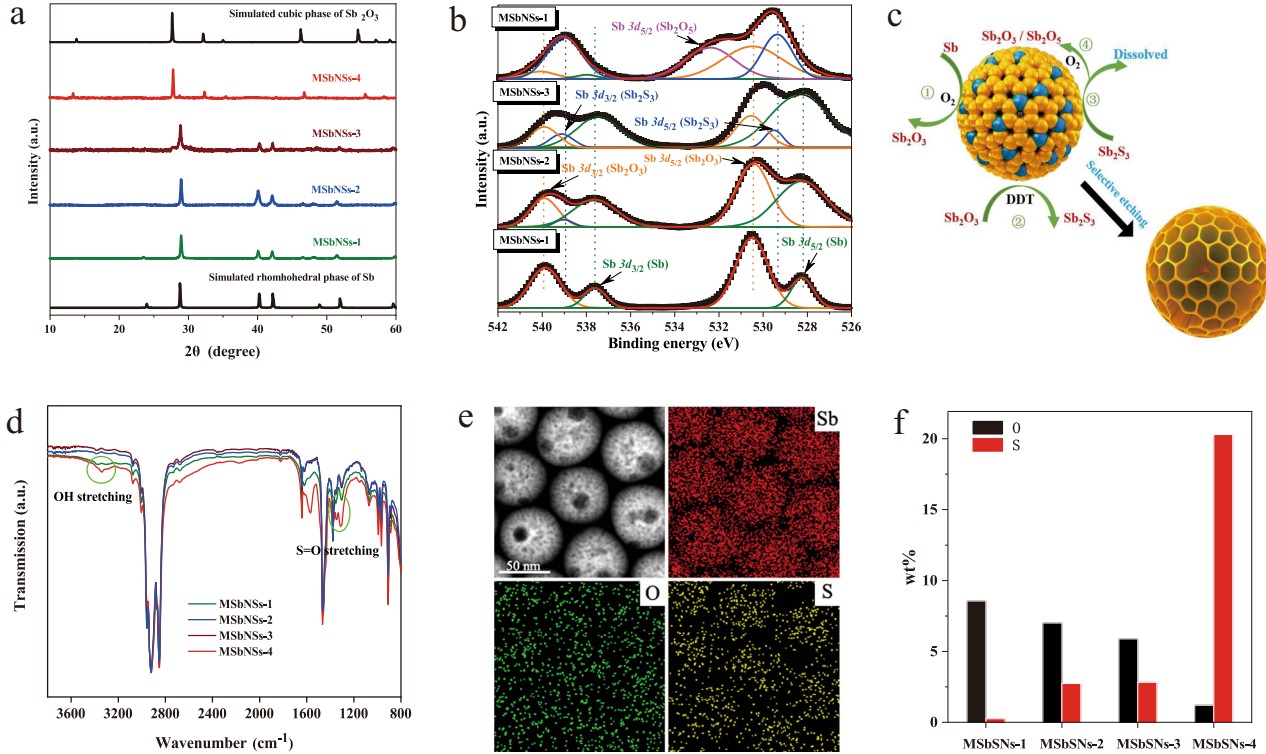

**Fig. 2 Formation mechanism study of MSbNSs. a** XRD patterns, **b** XPS analysis of MSbNSs-1/2/3/4. **c** Step-oxidization reaction mechanism of MSbNSs. **d** FTIR spectra of MSbNSs-1/2/3/4. **e** EDS mapping of MSbNSs-3. **f** The change of residual ratios of O and S elements in MSbNSs-1/2/3/4, respectively. Experiments above were performed three times with similar results.

experimental parameters were kept constant. As the concentration of DDT increased, the obtained MSbNSs showed obvious mesoporous structures and good monodispersity. When the volume of DDT reached 2.0 mL, the diameter of the obtained MSbNSs decreased significantly. Numerous small nanoparticles appeared, which was ascribed to the excessive etching of the mesoporous structures. The results demonstrated that high concentration of DDT can accelerate the etching process of mesoporous structures (Supplementary Fig. 7). Therefore, we proposed a step-oxidization reaction mechanism to form the mesoporous structure in the following three steps as shown in Fig. 2c.

$$Sb + O_2 \rightarrow Sb_2O_3 \quad (1)$$

$$Sb_2O_3 + C_{12}H_{25}SH \rightarrow Sb_2S_3 + C_{12}H_{25}OH \quad (2)$$

$$Sb_2S_3 + O_2 \rightarrow Sb_2O_3 + SO_2 \quad (3)$$

$$Sb_2O_3 + O_2 \rightarrow Sb_2O_5 \quad (4)$$

Step (2) and Step (3) were further confirmed by measuring the Fourier-transform infrared spectroscopy (FTIR) spectra of the reaction solutions (Fig. 2d). O-H stretching of alcohol (3550–3200 cm$^{-1}$) and S=O stretching of SO$_2$ (1350–1300 cm$^{-1}$) were tested to be stronger as the increase of O$_2$ amount, especially in the solvent of MSbNSs-4. Elemental mapping using energy-dispersive X-ray spectroscopy (EDS) under STEM mode further confirmed the existence of O and S in MSbNSs-3 (Fig. 2e, Supplementary Table 1). The corresponding mass percentages of Sb, O and S elements in MSbNSs-3 are 94.5%, 3.3% and 2.2%, respectively (Fig. 2f, Supplementary Fig. 8). From MSbNSs-2 to MSbNSs-4, although the amount of O$_2$ in the reaction precursor were increasing, the residual ratios of O elements in the final products were decreasing, while the ratios of S element were increasing with a significant

increase in MSbNSs-4, which further proved our proposed reaction mechanism.

**NIR-II photothermal, degradation and drug loading/release performance of MSbNSs.** Different from the good photothermal stability of Sb nanopolyhedrons reported in our previous work[37], the as-synthesized MSbNSs showed varying degrees of photothermal degradation under 1210 nm laser irradiation (Fig. 3a). In particular, MSbNSs-3 showed the most obvious photothermal degradation among MSbNSs-1/2/3. The detailed degradation behavior of MSbNSs-3 was studied by TEM (Fig. 3b). Obvious degradation of the mesopores in MSbNSs-3 and numerous soluble Sb species could be observed after two cycles of irradiation. Almost all the spherical structures were destroyed after three successive cycles of irradiation, indicating that MSbNSs-3 gradually collapsed after laser irradiation. The color of the black MSbNSs-3 solution also gradually faded to transparent after three cycles of irradiation. The photothermal stability of MSbNSs-2 decreased to some extent, but the degradation of MSbNSs-2 proceeded slowly (Supplementary Fig. 9). The degradation of the mesopores in MSbNSs-2 can also be observed after three successive cycles of irradiation, but the collapse of mesoporous nanostructures would not appear. By comparison, MSbNSs-1 showed the best photothermal stability (Supplementary Fig. 10). The photothermal degradation mechanism of MSbNSs was also analyzed based on the FDTD simulations as Sb-based nanostructures possess localized surface plasmon resonance (LSPR) effect[46]. The detailed fitting process is described as shown in supporting information. The results showed that electric fields at the joints and gaps inside the mesoporous structure were amplified by more than several orders of magnitude, resulting in stronger localization of heat generation in mesopores. The simulation results indicated that the small mesopores of both

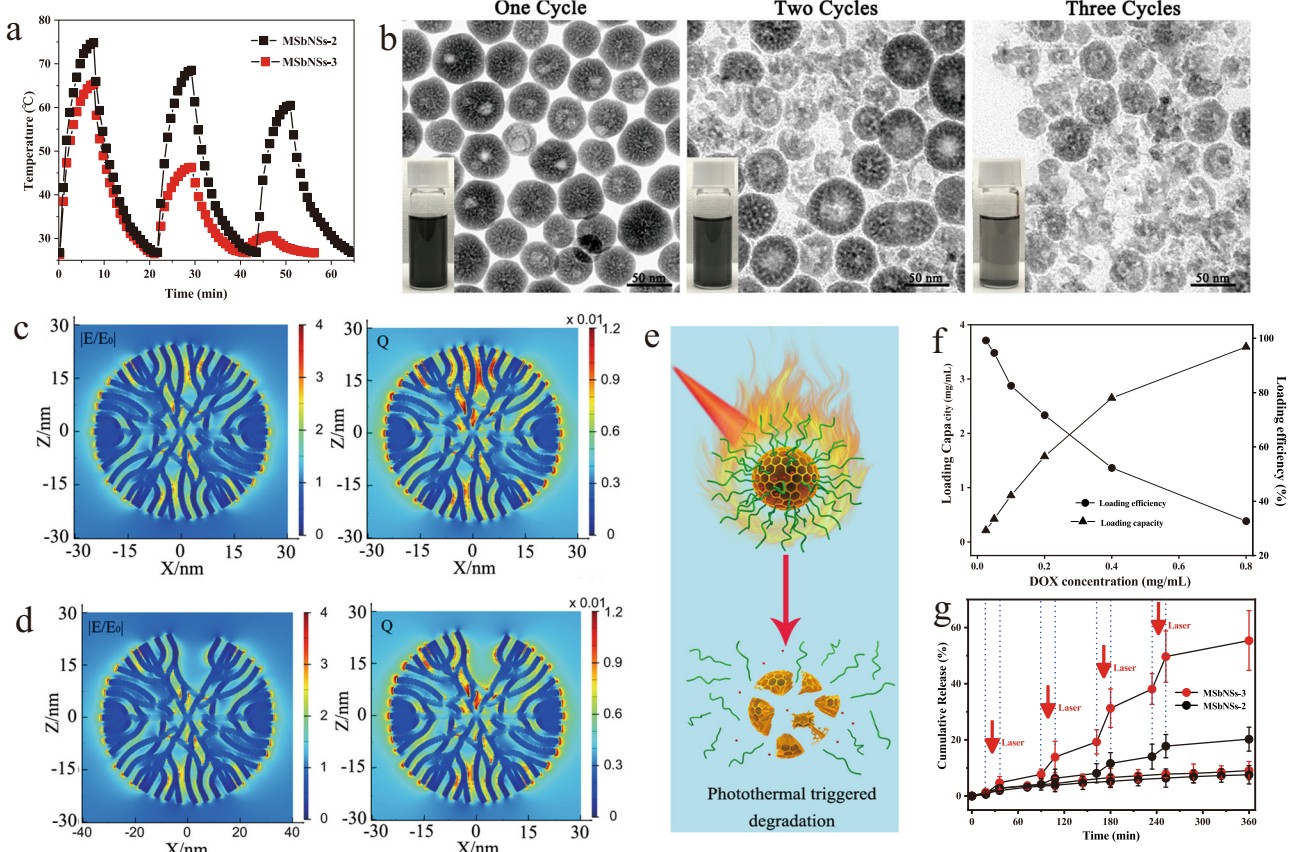

**Fig. 3 Photothermal degradation and drug loading/release of MSbNSs. a** Photothermal stability study of PEGylated MSbNSs-2/MSbNSs-3 under the irradiation of 1210 nm light. Experiments were performed three times with similar results. **b** TEM images of PEGylated MSbNSs-3 after different heating and cooling cycles. A representative image of 3 replicates from each group is shown. Simulation results of the electric field enhancement ($|E/E_0|$) and the heat power density Q inside the **c** MSbNSs-2 and **d** MSbNSs-3 illuminated by 1210 nm light. **e** Schematic illustration of photothermal triggered degradation and DOX release. **f** DOX loading capacity and efficiency of MSbNSs-3. Experiments were performed two times with similar results. **g** Photothermal triggered DOX release in PBS (pH = 7.4) irradiated with 1 W cm$^{-2}$ of a 1210 nm laser. Data are expressed as means ± SD ($n = 3$).

MSbNSs-2 (Fig. 3c) and MSbNSs-3 (Fig. 3d) could generate localized "hot channels" with enhanced electric field within the mesopores and similar heat power densities were generated. "Hot channels" would gradually disappear as mesopores grew larger (~11.5 nm). MSbNSs-3 have various mesopore sizes and larger mesopores would lead to fewer "hot channels" than MSbNSs-2, which is consistent with the change of the photothermal performance of MSbNSs-2 and MSbNSs-3. During the photothermal process, the increased temperature would promote the following reaction (Step (5)) of surface Sb atoms with water molecules to form Sb-H and Sb-OH, leading to the dissolution of Sb nanocrystals in aqueous solution[47].

$$Sb + 2H_2O \rightarrow Sb-OH_2 + Sb-HOH \rightarrow Sb-H + Sb-OH \tag{5}$$

Since MSbNSs-3 have larger pore sizes, the amount of accessible water within the NSs would be higher. The photothermal degradation process was further promoted in MSbNSs-3. The looser mesoporous structure and lower crystallinity of MSbNSs-3 could also accelerate the dissolution and result in the enhanced photothermal degradation property in MSbNSs-3.

The photothermal degradation property of MSbNSs can further boost the release efficiency as nanocarriers for different types of drugs. To check the distribution of DOX loaded within the mesopores of MSbNSs, the elemental mappings of Cl element

in DOX before and after loading were measured as shown in Supplementary Fig. 11, indicating that DOX can be uniformly loaded within MSbNSs. We evaluated their capabilities of drug delivery and NIR-II triggered on-demand release of drugs using DOX as a typical anticancer model drug (Fig. 3e). The drug loading capacity and efficiency and of MSbNSs-2 and MSbNs-3 were studied as shown in Fig. 3f. The performance of drug release at different pH values was also provided as shown in Supplementary Fig. 12. A burst in the release of DOX (~56%) can be observed with 1210 nm irradiation, which is about 6 times higher than that of the group without NIR-II light irradiation (Fig. 3g). It is noteworthy that the on-demand drug release of MSbNSs can also be observed under the irradiation of laser with other wavelength (Supplementary Fig. 13). The on-demand release properties under NIR-II laser irradiation could be ascribed to the collapse of the MSbNSs and local photothermal effect to accelerate the diffusion of drugs. All of the above results reveal that the MSbNSs can be used simultaneously as agents for PTT, PAI and drug carriers for NIR-II controlled drug release.

Considering the appropriate pore size and NIR-II absorption wavelength as well as photothermal degradation, we have chosen MSbNSs-3 to perform multimodal theranostics in the following work. MSbNSs-3 were PEGylated and the absorption properties of PEGylated MSbNSs-3 in aqueous solution still exhibited broad absorption from visible to NIR wavelengths and the strongest absorption peak was 1480 nm, located within the NIR-II window (Fig. 4a). The strong absorption of PEGylated MSbNSs-3 in the

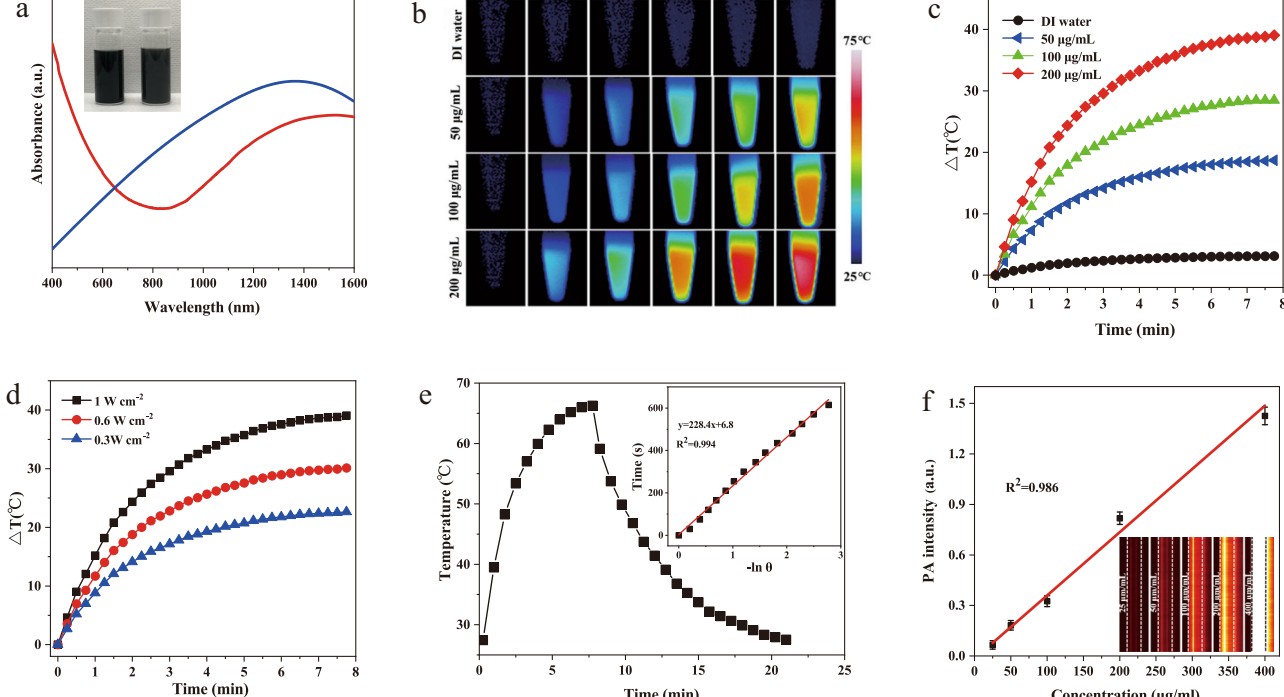

**Fig. 4 NIR-II photothermal performance of PEGylated MSbNSs-3. a** Vis-NIR spectra of MSbNSs-3 before (blue line) and after (red line) PEG modification. **b** Photothermal images and **c** temperature change curves of the PEGylated MSbNSs-3 with different concentrations under 1210 nm laser irradiation for about 8 min. **d** Temperature curves of the PEGylated MSbNSs-3 under 1210 nm laser irradiation at different power densities for about 8 min. **e** The photothermal heating curve of the PEGylated MSbNSs-3 under laser irradiation (1210 nm, 1 W cm$^{-2}$) and cooling curve after turning off the laser. The inset shows the linear relationship between –lnθ and time obtained from the cooling. Experiments above were performed three times with similar results. **f** Photoacoustic amplitudes at 1210 nm as a function of concentrations of the PEGylated MSbNSs-3. Data are expressed as means ± SD ($n = 3$).

NIR-II region suggest their potential in NIR-II PAI and PTT. The photothermal effect of PEGylated MSbNSs-3 was studied by the irradiation of 1210 nm light. This provides another efficient NIR-II excitation wavelength besides the typical 1064 nm. As shown in Fig. 4b, c, the temperature variation of PEGylated MSbNSs-3 displayed a concentration-dependent behavior. The photothermal properties of MSbNSs-3 were also studied by varying the excitation wavelength (Supplementary Fig. 14). The results demonstrated that MSbNSs-3 showed excellent photothermal performance under the irradiation of 1064 nm laser, while the temperature increase under 1064 excitation was slightly lower than that under 1210 nm excitation. We also investigated the photothermal performances of PEGylated MSbNSs-3 irradiated under different power densities (Fig. 4d). The PTCE of PEGylated MSbNSs-3 was calculated to be ~39% (Fig. 4e), which is superior to the majority of current NIR-II PTAs. By comparison, PEGylated MSbNSs-1 and MSbNSs-2 was calculated to be ~44% and ~41% (Supplementary Fig. 15). Moreover, a strong concentration-related photoacoustic signal of PEGylated MSbNSs-3 was observed (Fig. 4f), implying that PEGylated MSbNSs-3 could be used as bright NIR-II photoacoustic contrast agents.

**In vitro synergistic PTT and chemotherapy.** To study the synergistic therapeutic outcomes of DOX-loaded PEGylated MSbNSs-3 (PEGylated MSbNSs-3/DOX), the cellular internalization process was checked in panc02 cancer cells using confocal microscopy. The cellular uptake of PEGylated MSbNSs-3/DOX in panco2 cells after incubation was checked by confocal fluorescence (Fig. 5a) and flow cytometry analysis (Supplementary Fig. 16). The fluorescence of DOX overlapped with the fluorescence of Lyso-tracker, indicating that the uptake process was mainly achieved by endocytosis. The cytotoxicity and treatment efficacy of PEGylated MSbNSs-3/DOX were further tested by Live/dead assays. PEGylated MSbNSs-

3 with/without DOX were incubated with panc02 cancer cells under different concentrations from 0 to 100 μg mL$^{-1}$ (Fig. 5b). Without 1210 nm irradiation, PEGylated MSbNSs-3 showed a high cell viability indicating a negligible cytotoxicity, meanwhile PEGylated MSbNSs-3/DOX would kill cancer cells by releasing DOX. The in vitro chemotherapy efficacy is limited compared with NIR-II PTT. The 1210 nm irradiation on PEGylated MSbNSs-3 could generate heat to kill cancer cells, leading to a cell viability reduced to 20% at 100 μg mL$^{-1}$. The synergistic PTT and chemotherapy showed an improved in vitro therapeutic capability with a minimal cell viability of 8% (Supplementary Fig. 17). The ex vivo fluorescence imaging showed that the accumulation of PEGylated MSbNSs-3 in tumor was markedly higher than that in other organs at 8 h post-treatment. This data indicated that PEGylated MSbNSs-3 had excellent tumor accumulation efficiency (Supplementary Fig. 18). The time-dependent biodistribution of PEGylated MSbNSs-3 in mice without laser irradiation was further studied as shown in Supplementary Fig. 19, which showed that PEGylated MSbNSs-3 were mostly accumulated in tumor, liver, spleen, and kidney.

**In vivo NIR-II PAI-guided PTT and chemotherapy.** In vivo 1210 nm PAI was conducted to guide the optimal systemic administration window of PEGylated MSbNSs-3/DOX. The PA signal in the tumor increased gradually after the intravenous injection of PEGylated MSbNSs-3/DOX, indicating the gradual accumulation of these nanomedicines in the tumor site (Fig. 5c). The PA signal achieved the highest at 8 h post-injection, with a 9-fold higher PA amplitude than the pre-injection tumor (Supplementary Fig. 20). With the guidance of PAI, the photo-irradiation of 1210 nm laser was conducted at 8 h post-injection to prove the therapeutic efficacy of the synergistic NIR-II PTT and chemotherapy on panc 02 tumor-bearing mice. The tumor

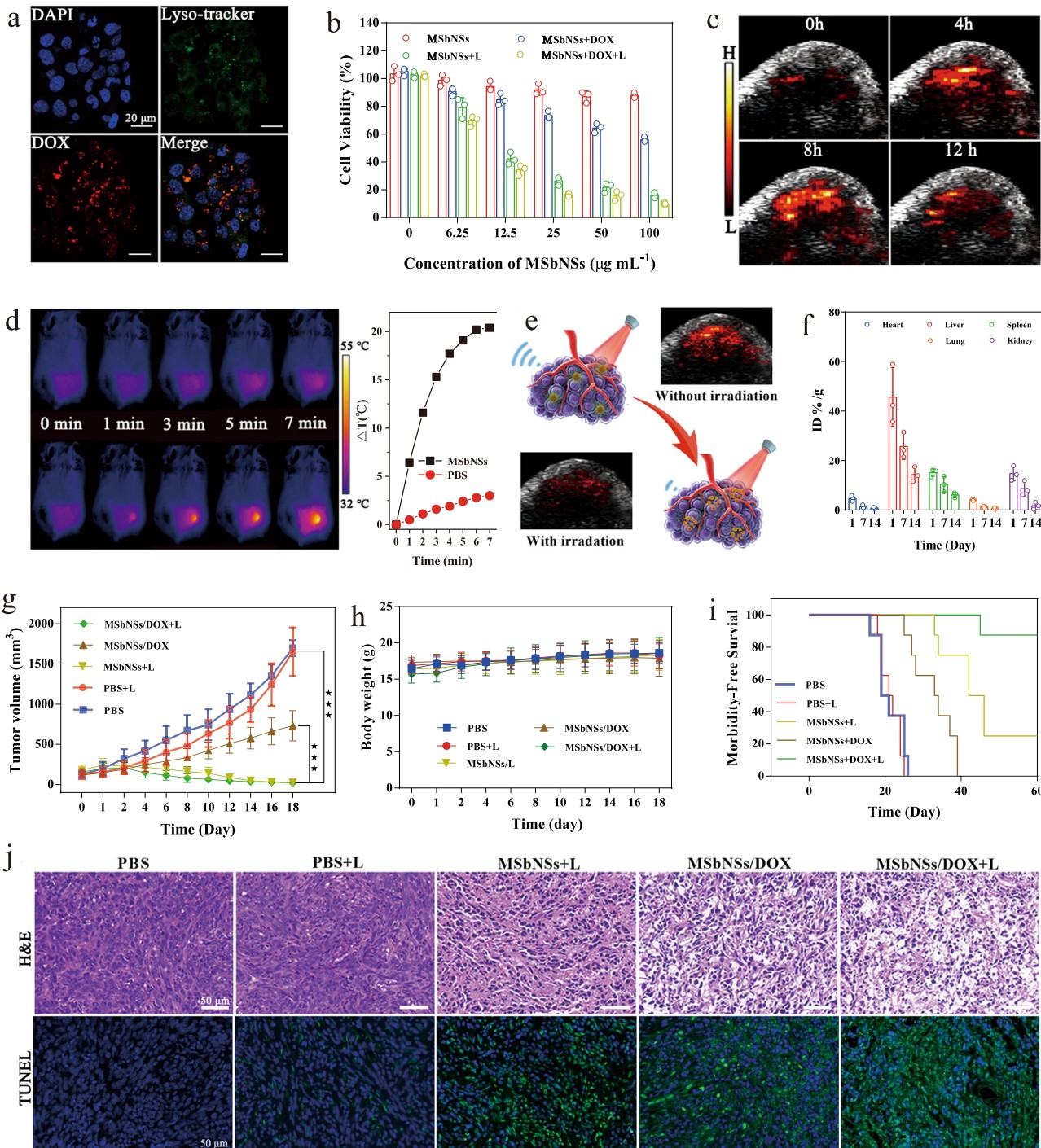

**Fig. 5 In vitro and in vivo synergistic chemotherapy and NIR-II PTT via PEGylated MSbNSs-3/DOX. a** Confocal microscopy images of panc02 cancer cells to show the internalization of PEGylated MSbNSs-3/DOX. **b** Cell viability of panc02 cells treated with PEGylated MSbNSs-3, PEGylated MSbNSs-3 + laser, PEGylated MSbNSs/DOX, and PEGylated MSbNSs-3/DOX + laser. Data are expressed as means ± SD ($n = 3$). **c** PA images showing changes in PA signal (red) over time with PEGylated MSbNSs-3/DOX being injected at the tumor site. A representative image of two individual mice per group is shown. **d** IR thermal images of panc02-tumor mice injected with PBS and PEGylated MSbNSs-3 in response to 1210 laser irradiation. A representative image of three individual mice per group is shown. **e** Schematic illustration of in vivo photothermal triggered degradation and the change of PA images before and after irradiation for 15 min. **f** Biodistribution of PEGylated MSbNSs-3 using ICP to measure the concentration of Sb element in major organs. Data are expressed as means ± SD ($n = 3$). **g** Tumor growth curves. P-values were calculated by Student's two-sided t-test. (***$P < 0.001$), **h** body weight, and **i** survival proportions of mice after systemic treatment with PBS, PBS + laser, PEGylated MSbNSs-3 + laser, PEGylated MSbNSs-3/DOX, and PEGylated MSbNSs-3/DOX + laser. Data are expressed as means ± SD ($n = 8$). **j** H&E and TUNEL staining for tumor acquired 24 h after systemic treatment. A representative image of 3 individual mice per group is shown.

temperature of PEGylated MSbNSs-3/DOX treated mice increased rapidly to ~46 °C within 3 min and finally arrived to ~50 °C at 7 min, which is significantly higher than that of PBS-treated tumors (Fig. 5d). To further investigate the in vivo photothermal triggered degradation of PEGylated MSbNSs-3, PA images before and after laser irradiation were studied as shown in Fig. 5e. An obvious photothermal degradation of PEGylated MSbNSs-3 was observed after the irradiation of 1210 nm laser for 10 min, which was indicated by the dramatic drop of PA intensity of tumor by 70% (Supplementary Fig. 21). To assess the clearance of PEGylated MSbNSs-3, the biodistribution of MSbNSs-3 in mice was studied using inductively coupled plasma (ICP) to measure Sb element in major organs. After the PEGylated MSbNSs-3 were injected into mice and with irradiation of 1210 nm laser, the distribution of PEGylated MSbNSs-3 in major organs were measured at 1, 7, 14 days. PEGylated MSbNSs-3 were mainly cumulated in the liver, spleen and kidney (Fig. 5f).

The tumor sizes in different groups were measured and compared (Fig. 5g). Mono chemotherapy by PEGylated MSbNSs-3/DOX showed a mild tumor inhibition rate, consistent with former results using mesoporous silica structures as nanocarriers[48,49]. Meanwhile, the NIR-II PTT by PEGylated MSbNSs-3 showed excellent therapeutic outcomes with eradication of tumors owing to the excellent photothermal performance of MSbNSs-3. Tumors treated by PEGylated MSbNSs-3/DOX for synergistic NIR-II PTT and chemotherapy were also eradicated, with an even faster speed. The body weights of mice in all the groups were not influenced by different treatments in all the groups (Fig. 5h). The survival rates of mice were monitored to further prove the treatment efficacy. The treatment of NIR-II PTT and chemotherapy resulted in a highest survival rate (Fig. 5i). Moreover, there were also no obvious histopathological abnormalities found in major organs of the mice (Supplementary Fig. 22), indicating the good biocompatibility of the MSbNSs nanoplatforms. Also, the results of hematology analysis and blood biochemical test indicated PEGylated MSbNSs-3 had no obvious long-term side effect in vivo (Supplementary Fig. 23). Hematoxylin and eosin (H&E) staining and TdT-mediated Dutp-biotin nick end-labeling (TUNEL) staining of tumor slices evaluated the necrosis and apoptosis of tumor cells in all the groups (Fig. 5j). No apparent necrosis or apoptosis was found in control groups, PEGylated MSbNSs-3/DOX caused moderate necrosis and apoptosis by releasing DOX. Large areas of necrosis and apoptosis were found in 1210 nm irradiated PEGylated MSbNSs-3 and PEGylated MSbNSs-3/DOX groups. The synergistic NIR-II PTT and chemotherapy caused the most necrosis and especially apoptosis, which is also consistent with the tumor growth curves. Altogether, these data demonstrated that our PEGylated MSbNSs-3/DOX nanoplatform showed high potential for multimodal NIR-II theranostics with therapeutic efficacy, good biocompatibility and photothermal degradability.

## Discussion
In summary, we developed a selective etching method to fabricate semimetallic mesoporous nanostructures with controllable mesopore size. By controlling the oxidative degree of Sb nuclei, the as-synthesized MSbNSs possessed mesopores with tunable pore sizes. The detailed selective etching reaction processes and mechanism were further explored and explained. Moreover, MSbNSs also demonstrated NIR-II photothermal performance, showing a strong NIR-II PAI contrast and relatively high NIR-II PTCE (~44%). Thus, MSbNSs have shown great potential as integrated nanoplatforms for NIR-II PAI guided synergistic chemotherapy and NIR-II PTT. We also observed that the enhanced photothermal degradability of MSbNSs as the sizes of mesopores increases. The FDTD calculation results revealed that

the etched mesopores would lead to localized "hot channels" within the mesopores, which would accelerate the degradation of Sb nanostructures. Larger pore sizes would lead to a higher amount of water within the MSbNSs, looser structure and lower crystallinity, leading to a faster photothermal degradation.

With the guidance of in vivo NIR-II PAI, the synergistic NIR-II PTT and chemotherapy by PEGylated MSbNSs-3/DOX showed therapeutic efficacy with tumor elimination and highest post-treatment survival rate. The multimodal theranostic nanoplatform could also provide various combinations of NIR-II PAI and PTT with other therapeutic modalities via loading various components. Our selective etching method can be widely applied in different material systems, including 2D materials, multilayer structures and composite materials etc. utilizing semimetals for enhanced performance and multifunctionality.

## Methods
**Materials**. 1,2-distearoyl-sn-glycero-3-phosphoethanolamineN-[amino(polyethylene glycol)2000] (DSPE-PEG) were purchased from Avanti Polar Lipids (USA). Dodecylthiol (DDT, 99.9%), Oleylamine (OLA, 70%), Oleic acid (OA, 90%), Technical grade octadecene (ODE, 90%), tert-Butylamine borane (99%), and antimony trichloride (SbCl3, 99.999%) were purchased from Sigma Aldrich. Toluene and absolute ethanol were purchased from Beijing Chemical Reagent Ltd., China. Unless otherwise mentioned, all chemicals were used as received without further purification.

**Cells and animals**. The murine pancreatic cancer cell Panc02 was purchased from cell bank of Chinese Academy of Science, Shanghai. Cells were cultured in DMEM supplemented with 10% fetal bovine serum and 1% penicillin-streptomycin solution at 37 °C and 5% CO2 in a humidified atmosphere. Female C57BL/6 mice (6 weeks) were purchased from Guangdong Medical Laboratory Animal Center (GMLAC, Guangzhou, China). All mice were housed under specific pathogen-free conditions at 18–23 °C, and 40–60% humidity with free access to food and water. All the animal experimental procedures were performed in accordance with the Shenzhen University Medical Laboratory Animal Center regulations and approved protocols. The maximal tumor size must not exceed 2.0 cm permitted by ethics committee. Animals in our experiments were sacrificed when an aspect of the tumor lengths reached a size of 1.5 cm.

**Synthesis of MSbNSs with tunable pore sizes**. In a typical synthesis of MSbNSs, 1 mL Dodecylthiol (DDT), 5 mL Oleylamine (OLA), 3 mL octadecene (ODE) and antimony trichloride (SbCl3) (0.05 g, 0.22 mmol) were sequentially added in a 50 mL three necked flask. Then the reaction mixture was stirred and degassed under vacuum at 50 °C until the SbCl3 dissolved completely in solvent. Next, the flask was heated to 170 °C. On the other hand, tert-Butylamine borane (0.03 g, 0.35 mmol) was dissolved completely in 1 mL of OLA. Then, 1 mL solution of tert-Butylamine borane in OLA was injected swiftly to the reaction system, simultaneously the pumping time of O2 from 1 s/10 s/30 s was performed for controlling the amount of O2 in the reaction precursor. After stirring for 30 min, the reaction was stopped by immersing the flask into the ice-water bath. At 50 °C, dried OA (0.1 mL) was added to displace weakly bound OLA ligand capping. Sb NPHs were precipitated by ethanol 15 mL, separated by centrifugation, redispersed in toluene (5 mL) containing OA (0.1 mL). The samples were sealed and kept in dark before using. The morphology of MSbNSs can be tuned by adjusting the mole of SbCl3 and mole ratio of DDT/OLA.

**Synthesis of PEGylated MSbNSs**. The as-synthesized MSbNSs were dispersed in dichloromethane (5 mL, 4 mg mL⁻¹), followed by mixing with 1,2- distearoyl- sn-glycero-3- phosphoethanolamineN-[amino(polyethylene glycol)2000] (DSPE-PEG2K) at concentrations at 4 mg mL⁻¹. Subsequently, the resulting mixture was sonicated for about 30 min, and the solvent of dichloromethane was fully removed via rotary evaporator. The residual solid was redispersed by adding 4 mL purified water and sonicated for 30 min. The resulting products were collected by centrifugation 10000 rpm for 5 min, washed with purified water three times, and redispersed in 4 mL PBS. The samples were sealed and kept in dark before using.

**DOX loading and releasing**. DOX were added into the resultant PEGylated MSbNSs-3 solution (1 mg mL⁻¹). After magnetically stirring for 24 h, unloaded drugs were removed by centrifugation at 8000 rpm. To determine the drug loading capacity and efficiency of the nanoparticles. Different concentrations of DOX (0~0.8 mg mL⁻¹ of DOX) were stirred with PEGylated MSbNSs-3 (0.2 mg/mL) for 24 at room temperature. The resultant product was collected by centrifugation at 8000 rpm. The amount of unbound DOX in the solution was determined by measuring the absorbance at 488 nm using a calibration curve prepared under the same condition. The drug loading capacity is defined as (weight of drug loaded on

PEGylated MSbNSs-3/weight of PEGylated MSbNSs-3). The drug loading efficiency (%) is defined as (weight of drug loaded on PEGylated MSbNSs-3/weight of initially added DOX) × 100.

**Cellular uptake**. The cellular uptake of the PEGylated MSbNSs/DOX was observed using a confocal laser scanning microscope. In brief, panc 02 cells were seeded into 8-well chambered slides (Thermo Scientific, USA) and cultured for 12 h. Then, the cells were incubated with PEGylated MSbNSs/DOX for 3 h, and followed by Lyso-Tracker (green color) and Hoechst 33342 (blue color) double staining. The cells were washed with PBS for three times and analyzed via confocal laser scanning microscope.

**In vitro phototoxicity of PEGylated MSbNSs**. CCK-8 assay was used to evaluate the phototoxicity of PEGylated MSbNSs. Briefly, panc 02 cells were seeded into a 96-well plate with the concentration of $5 \times 10^3$ cells per well and cultured for 12 h. Then, the medium was replaced with serum-free medium containing PEGylated MSbNSs-3, PEGylated MSbNSs-3/DOX of various concentrations for another 16 h, respectively. Then, each well was washed three time with PBS to remove free MSbNSs. The cells were submitted to 1210 nm laser irradiation at 0.75 W cm$^{-2}$ for 5 min and then culture medium was added to each well to incubate for 24 h. The amount of an orange formazan dye, produced by the reduction of WST-8 (active gradient in CCK-8) by dehydrogenases in living cells, is directly proportional to the quantity of living cells in the well. Therefore, by measuring the absorbance of each well at 450 nm using a microplate reader, cell viability could be determined by calculating the ratio of absorbance of experimental wells to that of the control cell wells.

**PA imaging**. An optical parametric oscillator laser source (Innolas GmbH, Bonn, Germany) emitting 8 ns pulsed lasers was coupled to a multimode optical fiber with a 1500 μm core diameter for the photoacoustic signal excitation. The use of this nanosecond pulsed laser on the tumor tissue resulted in very small thermal expansions. This led to the vibration of the tissue, which generated ultrasonic waves that could be recorded by a commercial 128-element linear array transrectal ultrasound transducer. The Q-switch output of the laser source was synchronized with a Vantage 128 research ultrasound platform (Verasonics, Inc. Kirkland WA, USA) to perform photoacoustic and ultrasound data acquisition. Photoacoustic signals were acquired at a frame rate equal to the laser pulse firing frequency of 30 Hz, and the conventional delay and sum (DAS) reconstruction algorithm was used to reconstruct the photoacoustic B-scan image. The system can simultaneously display both photoacoustic and ultrasound images in real-time.

**Mouse tumor model**. All of our animal experiments were conducted in compliance with the guidelines established by the Animal Ethical and Welfare Committee of Shenzhen University (AEWC-SZU). For photothermal therapy, the tumors were established on female C57BL/6 mice by subcutaneous injection of two million panc02 cells suspended in 0.2 mL of DMEM (10% FBS, 1% antibiotics) into side of the back of mice. Tumors were allowed to grow until a single aspect was longer than 6 mm before being used for photothermal treatment.

**In vivo Photothermal therapy of PEGylated MSbNSs**. After tumor volume reached 100–150 mm$^3$, mice were randomly divided into 5 groups (8 mice per group): PBS, PBS+L, PEGylated MSbNSs-3/DOX, PEGylated MSbNSs-3+L, and PEGylated MSbNSs/DOX+L. At 8 h post-injection, the tumor regions of living mice were exposed to a 1210 nm laser with a power density of 1 W/cm$^2$, using an optical fiber with a diffusion lens (Pioneer Optics, Bloomfield, CT) to delivery uniform light distribution on the treatment surface. The temperature on the tumor surface during PTT was measured with an infrared thermal camera E80 (FLIR, Boston, USA). After photothermal treatment, sizes of tumors were measured by a caliper every other day for 18 days. Tumor volume was calculated as follows: volume = (tumor length) × (tumor width)$^2$/2. One day after treatment, tumors were collected from selected mice in each group were collected for tissue section and H&E staining.

**Characterizations**. TEM and STEM observations were taken on a FEI Titan Cubed Themis G2300 transmission electron microscope with an acceleration voltage of 300 Kv. Carbon coated copper grids were dipped in the chloroform solutions to deposit the samples onto the films. UV-visible spectra were recorded on an HP8453 UV-visible spectrophotometer. X-ray powder diffraction (XRD) patterns were obtained by using a Philips pw1830 X-ray diffractometer. X-ray photoelectron spectroscopy (XPS) were obtained by using K-Alpha+. Dynamic light scattering (DLS) measurement was performed by NanoBrook ZetaPlus. Confocal images were taken using a Leica TSC SP8 confocal laser scanning microscope (Germany) with a 63× oil-immersion objective.

**Calculation of the photothermal conversion efficiency**. The photothermal conversion efficiency of MSbNSs was determined according to a previously reported method as follows Equation[50].

$$\eta = \frac{hA\Delta T - Q_s}{I\left(1 - 10^{-A_\lambda}\right)}$$

where $h$ is the coefficient of heat transfer, A is the container surface area, $\Delta T$ is the temperature change of the PEGylated MSbNSs-3 solution, $I$ is the power density of the NIR laser, $A_\lambda$ is the absorbance of the solution of PEGylated MSbNSs-3 at 1210 nm, and $Q_s$ is the heat associated with the light absorbance of the water.

**FDTD Calculations**. Herein, to confirm the spatial electric field distribution of as-prepared MSbNSs under the irradiation of a beam of linearly polarized light, the finite-difference-time-domain (FDTD) simulation (8.11.337 version, Lumerical Solutions, Inc.) was performed as an effective approach. The complex refractive indexes of Sb was adopted from the book "The Handbook on Optical Constants of Metals: In Tables and Figures" edited by Sadao Adachi. *The handbook on optical constants of metals: in tables and figures. World Scientific.* The complex refractive indexes of the media H$_2$O was adopted from the simulation database "H$_2$O (water) - Palik" All geometric parameters for simulations were consistent with the average actual size of as-prepared samples shown in TEM image. In the FDTD simulations, the diameter and pore diameter of MSbNSs-2 were set as 55 nm and 3.7 nm, while MSbNSs-3 were set as 55 nm and 3.7 nm/11.5 nm. For the detailed FDTD parameter setting, the simulation region was set as a unit 700 nm × 700 nm × 700 nm in 3Ds. Herein, boundary conditions were set as perfectly matched layers (PML) in all simulations. To save computational resources while improving computational accuracy, a refined mesh grid near the structure was set as 0.3 nm in a 3D dimension 70 nm × 70 nm × 70 nm. Total-field scattered-field (TFSF) linearly polarized light waves, which was polarized in line with X-axis with a wavelength range of 400–1400 nm, were injected into the unit cell along the -Z direction. The amplitude of the electric field vector of the TFSF linearly polarized light vector was set to be 1 V m$^{-1}$. Two frequency-domain field profile monitors and two frequency-domain field and power monitors were localized at Z = 0 nm in the x-y plane and Y = 0 nm in the x-z plane, respectively. All the frequency-domain field profile monitors were utilized to collect electric field profile data of the surface and interior of and MSbNSs while keeping the excitation wavelength 835 nm and 1200 nm, respectively. Furthermore, the frequency-domain field and power monitors were utilized to collect the absorption power profile data of and MSbNSs under the excitation wavelength 1200 nm.

**Data analysis**. The statistical analysis of data was performed on Microsoft Excel 2010, Origin 9.1 and GraphPad Prism 7. For all statistical analysis, $**P < 0.05$ was regarded as statistically significant.

**Reporting summary**. Further information on research design is available in the Nature Research Reporting Summary linked to this article.

## Data availability
All data supporting the findings of this study are available within the Article, Supplementary Information or Source Data file. Source data are provided with this paper.

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

## Acknowledgements

This work was partially supported by the National Natural Science Foundation of China (61620106016/ 61835009/ 61775145/62005172/ 61805158/62127819); Guangdong Province Key Area R&D Program (2019B110233004); and Shenzhen Fundamental Research Project (JCYJ20190808160207366, JCYJ20190808114609361). The authors wish to acknowledge the assistance on HRTEM observation received from the Electron Microscope Center of the Shenzhen University. All animal experiments were performed under the regulations of Animal Ethical and Welfare Committee of Shenzhen University (AEWC-SZU).

## Author contributions

J.L.Q. and J.S. guided the overall research project. Y.C. conceived the idea, developed the mesoporous antimony nanospheres growth method, and carried out most of experiments and measurements. Z.Y. was involved in the analysis of all data and co-wrote the manuscript. Y.G.R. and C.B.L. contributes to the photoacoustic image. M.W., Q.W., Z.F.F., L.W.L. analyzed the data in part. K.Z. contributes to the FDTD measurements. All authors have approved the final version of the manuscript.

## Competing interests

The authors declare no competing interests.
