## [Peer Review File · Nature Communications]

Reviewers' Comments:

Reviewer #1:

Remarks to the Author:

Qu and co-authors reported a strategy for preparing monodispersed mesoporous antimony nanospheres (MSbNSs) through a partial oxidation of Sb nuclei and the selective etching of the as-formed Sb₂O₃. By changing reaction conditions, MSbNSs with different nanostructures were obtained with varying near-infrared absorption, which was suitable for near-infrared laser-based cancer theranostics. In addition, the yielded pores were used for drug delivery of DOX after laser-induced biodegradation of MSbNSs.

Overall, the manuscript focused on the synthesis and biomedical application of mesoporous antimony nanospheres, which was interesting but lack of novelty. For example, the liability of Sb to be oxidized in Sb-based nanomaterials for cancer treatment has been recently reported (Duo, Y. et al. *Adv. Funct. Mater.* 2020, 30, 1906010), the laser-induced biodegradation of Sb nanoparticles has been reported in other previous work (Ref. 40), and the NIR-II photothermal performance with a NIR-II PAI contrast and relatively high NIR-II PTCE (~44%) have already been reported by their own (Ref. 37) or other reports (Ref. 28, 39 and 40). This work presented a good combination based on mesoporous nanostructures. Also I found that the key advance of this work is tuning the mesoporous nanospheres with selective etching, however, the mechanism was not well supported and less convincing, and the controlling seemed to be not fine. Therefore, this work is interesting but lack of solid data to support the mechanism, considering the similar performance, it might be suitable to be published in a more specialized journal focusing on biomedical applications.

There are some other issues below that should be responded to before resubmission to this or another journal.

1. In Fig. 1f, the N₂ physisorption isotherms had too few dots, the determination of pore sizes with merely three dots was insufficient, and the line might be too flat to conclude an 18 nm pore structure in MSbNSs-4. The authors are encouraged to repeat these characterizations to avoid false signals of the hysteresis loop.
2. Different from the big differences in pore sizes or nanostructures, why the Brunauer-Emmett-Teller (BET) surface area of MSbNS-2, -3 and -4 (231, 235 and 234 m²g⁻¹) were almost the same and extensively higher than that of MSbNS-1 (78 m²g⁻¹). while the sizes of MSbNS-1, -2, -3 and -4 increased from 45, to 48, 51, and to final 58 nm. Without reasonable explanation or deep investigation, the phenomenon was more likely a summary of the experimental data, other than a mechanism with controllable tuning.
3. In Fig. S2, the Gaussian distribution of the average diameters of MSbNSs-4 with a final 58 nm was wrong.
4. The author mentioned that "The shape of MSbNSs can also be facilely tuned by regulating the reaction temperature (Supplementary Fig. 4)". Please provide reaction temperature and other parameters. Why the outer layer of these nanoparticles had no pores while there were some inside the nanoparticles, which were completely different from that of MSbNS-1, -2, -3 and -4.
5. Slight difference of Vis-NIR absorbance of nanoparticles with either blue or red-shifts is common, because the repeating synthesis of nanoparticles were not completely the same. In Fig. 1g, the difference of MSbNS-2 and -3 had little difference, the authors should repeat MSbNS-2 and MSbNS-3 for 5-10 times and then compare the difference of their absorbance to conclude the correlation in tuning red-shifts by tuning nanostructures. For comparison, the Vis-NIR absorption spectrum of solid MSbNSs should also be measured.
6. The authors mentioned "The collapsed MSbNSs-4 lost the strong NIR-II absorption, which is not applicable for the multimodal theranostics.", why?
7. It seemed that in this work a high absorbance in the second near-infrared region was desired, thus the measurement and comparison of the particular absorbance value or eat the same

concentrations of MSbNSs are essential.

8. The "step-oxidation reaction mechanism" proposed was not solid currently. As the cubic phase of Sb_2O_3 could be detected in MSbNSs-3 with a rather low peak, the proposed Sb_2S_3 should be detected as the reaction temperature was 170 °C, which should be high enough for crystallization.

9. Meanwhile, in Fig. 2e,f and Fig. S5, the EDS mapping results of MSbNSs-3 and -4 both showed high amount of S, XRD and XPS are strongly recommended to figure out whether the existence of Sb_2S_3 .

10. The claim on the four steps was also insufficient. For example, the authors said that "Step (2) and Step (3) were further confirmed by measuring the Fourier-transform infrared spectroscopy (FTIR) spectra of the reaction solutions (Fig. 2d). O-H stretching of alcohol (3550-3200 cm^{-1}) and S=O stretching of SO_2 (1350-1300 cm^{-1}) were tested to be stronger as the increase of O₂ amount, especially in the solvent of MSbNSs-4." In fact, DDT might also be oxidized by O₂ to generate O-H and S=O. To address this question, reactions without adding DDT should be performed.

11. Why the author chose 1210 nm to conduct photothermal therapy and laser-induced degradation, are other wavelengths of laser applicable?

12. In Fig. 3a and Fig. S7 and S8, the temperature values upon laser irradiation should be presented.

13. "Hot channel" is another proposed mechanism to distinguish the differences of MSbNSs-2 and -3. Could temperature values be determined in Fig. 3c,d to support the "stronger localization of heat generation" between different MSbNSs? More details would be better to explain the differences.

14. Why the PEGylation in Fig. 4a had such a big influence on the Vis-NIR absorption of MSbNSs with the strongest absorption peak moved to be 1480 nm?

15. Considering the liability of MSbNSs in vivo, the biodistribution and long-term toxicity should be studied.

16. The authors should mark all the figures with bars in Fig. 5a,j and Fig. S10 and S13.

Reviewer #2:

Remarks to the Author:

In this manuscript, a novel selective etching method to fabricate semimetallic mesoporous nanostructures was successfully established which could be used as efficient multimodal nanoplatforms for theranostics. Overall, this is a well-organized work, which can be considered for publication after the authors address the following issues.

1. It would be beneficial to further demonstrate the in vivo biodistribution of the MSbNSs.
2. The stability of MSbNSs in vivo should also be measured.
3. Some related references may be helpful to improve the introduction section of this manuscript.

Reply to reviewers

Manuscript ID: NCOMMS-21-18564

Manuscript Title: Degradable mesoporous semimetal antimony nanospheres for near-infrared II multimodal theranostics

Reviewers' comments:

Reviewer #1 (Remarks to the Author):

Qu and co-authors reported a strategy for preparing monodispersed mesoporous antimony nanospheres (MSbNSs) through a partial oxidation of Sb nuclei and the selective etching of the as-formed Sb_2O_3 . By changing reaction conditions, MSbNSs with different nanostructures were obtained with varying near-infrared absorption, which was suitable for near-infrared laser-based cancer theranostics. In addition, the yielded pores were used for drug delivery of DOX after laser-induced biodegradation of MSbNSs.

Overall, the manuscript focused on the synthesis and biomedical application of mesoporous antimony nanospheres, which was interesting but lack of novelty. For example, the liability of Sb to be oxidized in Sb-based nanomaterials for cancer treatment has been recently reported (Duo, Y. et al. Adv. Funct. Mater. 2020, 30, 1906010), the laser-induced biodegradation of Sb nanoparticles has been reported in other previous work (Ref. 40), and the NIR-II photothermal performance with a NIR-II PAI contrast and relatively high NIR-II PTCE (~44%) have already been reported by their own (Ref. 37) or other reports (Ref. 28, 39 and 40). This work presented a good combination based on mesoporous nanostructures. Also I found that the key advance of this work is tuning the mesoporous nanospheres with selective etching, however, the mechanism was not well supported and less convincing, and the controlling seemed to be not fine. Therefore, this work is interesting but lack of solid data to support the mechanism.

A: We appreciate Reviewer #1 for confirming the general interest and significance of our

manuscript. But for the comment of lack of novelty, **we sincerely encourage Reviewer #1 to re-compare the listed works with our manuscript.** The reference (Duo, Y. et al. Adv. Funct. Mater. 2020, 30, 1906010) that Reviewer #1 listed focused on the exploration of physiochemical properties of **2D antimonene** in biomedical applications that are not even related to mesoporous structure, while our work focused on the development of a new kind of **mesoporous antimony (Sb) nanospheres (MSbNSs)**. Development of mesoporous nanomaterials with diverse compositions and tunable pore sizes has always been an attracting but challenging issues in the past decade. **Therefore, we believed that the reported selective oxidation method for the preparation of MSbNSs is absolutely a major advance in the field of the preparation and design of multifunctional mesoporous nanomaterials.** The reference (Ref. 40) that Reviewer #1 listed provided a brief introduction of the photothermal degradability of Sb nanoparticles. In our work, we have studied both experimentally and theoretically in photothermal degradability of MSbNSs. Furthermore, **this is the first report of the fabrication of degradable MSbNSs and the potential multifunctional nanocarriers for theranostics.** The as-synthesized MSbNSs successfully combined the attractive physiochemical properties of Sb and its mesoporous architecture-dependent merits, which will certainly attract increasing attentions of mesoporous semimetal nanocarriers in multimodal nanoplatforms for theranostics. Therefore, the originality and novelty of our manuscript are indubitable. Our rebuttal is listed below one-by-one.

(1) *The liability of Sb to be oxidized in Sb-based nanomaterials for cancer treatment has been recently reported (Duo, Y. et al. Adv. Funct. Mater. 2020, 30, 1906010).* Duo, Y. et al. reported the potential biomedical applications of 2D antimonene, which is totally different from MSbNSs reported by our work. The oxidation is induced by X-ray and not related to structure design. On the other hand, the obtained antimonene nanomaterials were exfoliated by the ultrasonic from bulk antimony, which is distinct from our selective oxidation. The morphology of Sb nanomaterials obtained from ultrasonication is uncontrollable. Our group is one of the pioneers to prepare antimonene by ultrasonication, with a series publication in Angew. Chem. (2018, 130, 8804-8809; 2019, 58, 1574-1584) and Adv. Mater. (2018, 30, 1803244). **Compared with the ultrasonication approach, solution-phase synthesis of Sb nanomaterials with**

controllable morphology has great scientific importance and practical significance for biomedical applications. Therefore, we were committed to develop ligand-guided strategy for preparing high-quality Sb nanomaterials (Angew. Chemie, 2019, 131, 9996-10001; Adv. Mater. 2021, 33, 2100039). The selective oxidation method in this manuscript is novel and presents a major advance in the field of mesoporous semimetal nanomaterials.

(2) *The laser-induced biodegradation of Sb nanoparticles has been reported in other previous work (Ref. 40), but the biodegradation of Sb mesoporous nanospheres and its potential in on-demand drug release for theranostics has not yet been reported. Furthermore, we have studied both experimentally and theoretically in photothermal degradability of MSbNSs, which will certainly attract increasing attentions of mesoporous semimetal nanocarriers in multimodal nanoplatforms for theranostics.*

(3) *The NIR-II photothermal performance with a NIR-II PAI contrast and relatively high NIR-II PTCE (~44%) have already been reported by their own (Ref. 37) or other reports (Ref. 28, 39 and 40). Our work published in Adv. Mater. (Ref. 37) used Sb nanopolyhedrons and the excitation is within NIR-I range (808 nm). Ref. 28, 39 and 40 are all using Sb-based nanoparticles in the NIR-I range and their NIR-II performance has not been investigated, especially by 1210 nm (also the answer to Q. 11). Ref. 28 used Sb nanorods, Ref. 39 used antimonene quantum dots and Ref. 40 used antimonene nanoplates, none of these references even mentioned mesoporous structure.*

More discussions and details have been added in the revised manuscript according to Reviewer #1's kind suggestions to support the mechanism and clarify the concerns.

There are some other issues below that should be responded to before resubmission to this or another journal.

1. In Fig. 1f, the N₂ physisorption isotherms had too few dots, the determination of pore sizes with merely three dots was insufficient, and the line might be too flat to conclude an 18 nm pore structure in MSbNSs-4. The authors are encouraged to repeat these characterizations to

avoid false signals of the hysteresis loop.

A: We had a detailed communication with professional tester, and they were quite sure that the density of the test dot was big enough to reflect the trend of N₂ physisorption isotherms. In addition, the average pore sizes of MSbNSs were calculated by use of the BJH method, which has been mentioned in our manuscript. Some classic literatures were listed for your reference (*ACS Nano* 2010, 4, 529-539; *J. Am. Chem. Soc.* 2015, 137, 5903-5906; *Chem. Soc. Rev.*, 2013, 42, 3862-3875). The wide average distribution of pore size of MSbNS-4 is derived from collapsed pore structures. To show the results of pore sizes more clearly, the complete curves of pore sizes of MSbNSs-1/-2/-3 were provided separately as shown in Fig. S3.

2. Different from the big differences in pore sizes or nanostructures, why the Brunauer-Emmett-Teller (BET) surface area of MSbNS-2, -3 and -4 (231, 235 and 234 m²g⁻¹) were almost the same and extensively huger than that of MSbNS-1 (78 m²g⁻¹). while the sizes of MSbNS-1, -2, -3 and -4 increased from 45, to 48, 51, and to final 58 nm. Without reasonable explanation or deep investigation, the phenomenon was more likely a summary of the experimental data, other than a mechanism with controllable tuning

A: Thank you for your kind suggestion. We have repeated these characterizations for three times. The average surface area of MSbNS-2, -3, -4 were 252, 233, and 216 m²g⁻¹, respectively, which is consistent with the trend that the surface area decreases with the increase of the pore size (*J. Stat. Phys.* 1985, 38, 231; *Adv. Func. Mater.* 2020, 30, 2002725). The surface area of MSbNS-1 is significantly smaller than other samples owing to the insufficient etching structure.

3. In Fig. S2, the Gaussian distribution of the average diameters of MSbNSs-4 with a final 58 nm was wrong.

A: Thank you for your correction. We have corrected the Gaussian distribution by involving more MSbNSs in Fig. S2.

4. The author mentioned that “The shape of MSbNSs can also be facilely tuned by regulating the reaction temperature (Supplementary Fig. 4)”. Please provide reaction temperature and other parameters. Why the outer layer of these nanoparticles had no pores while there were some inside the nanoparticles, which were completely different from that of MSbNS-1, -2, -3 and -4.

A: Thank you for your kind suggestion. We have provided the reaction temperature and other parameters in the Methods and caption of Supplementary Fig. 5. When increasing the temperature of reaction, the shape of the products changed from sphere to polyhedron. The shape of polyhedron made it difficult to focus and show the whole mesoporous structure. To confirm the existence of mesopores at the outer layer, we provided new TEM and STEM images as shown in Fig. S5.

5. Sight difference of Vis-NIR absorbance of nanoparticles with either blue or red-shifts is common, because the repeating synthesis of nanoparticles were not completely the same. In Fig. 1g, the difference of MSbNS-2 and -3 had little difference, the authors should repeat MSbNS-2 and MSbNS-3 for 5-10 times and then compare the difference of their absorbance to conclude the correlation in tuning red-shifts by tuning nanostructures. For comparison, the Vis-NIR absorption spectrum of solid MSbNSs should also be measured.

A: Thank you for your kind suggestion. The strongest absorption peak of MSbNSs-2 is ~1300 nm, while the strongest absorption peak of MSbNSs-3 is ~1475 nm. We have repeated the MSbNSs-2 and MSbNSs-3 for several times. Although the vis-NIR absorbance of the as-synthesized MSbNSs were not completely the same, the strongest absorption peak of MSbNSs had a red-shift with the increase of pore size. To observe the change of absorption spectra clearly, the normalized absorption spectra of SbNSs/ MSbNS-1/ MSbNS-2/ MSbNS-3 were provided as shown in Fig. S6a. The absorption spectra of solid Sb nanoparticles were also added.

6. The authors mentioned “The collapsed MSbNSs-4 lost the strong NIR-II absorption, which is not applicable for the multimodal theranostics.”, why?

A: Thank you for your concern. The collapsed MSbNSs-4 are mainly composed of Sb_2O_3 confirmed by XRD and XPS, which has negligible photothermal conversion efficiency and does not have NIR-II absorption.

7. It seemed that in this work a high absorbance in the second near-infrared region was desired, thus the measurement and comparison of the particular absorbance value or eat the same concentrations of MSbNSs are essential.

A: Thank you for your concern. We have provided the comparison of the absorption of different MSbNSs at the same concentrations in Fig. S6b.

8. The “step-oxidization reaction mechanism” proposed was not solid currently. As the cubic phase of Sb_2O_3 could be detected in MSbNSs-3 with a rather low peak, the proposed Sb_2S_3 should be detected as the reaction temperature was 170 °C, which should be high enough for crystallization.

A: Thank you for your concern. The crystallization temperature of Sb_2S_3 is at least above 240 °C. Sb_2S_3 will remain amorphous at 170 °C and thus cannot be detected by XRD, we have added more discussions and explanations in the manuscript with references.

9. Meanwhile, in Fig. 2e,f and Fig. S5, the EDS mapping results of MSbNSs-3 and -4 both showed high amount of S, XRD and XPS are strongly recommended to figure out whether the existence of Sb_2S_3 .

A: Thank you for your concern. XRD and XPS were provided as shown in Fig. 2a and Fig. 2b. Although XPS analysis has proven the existence of Sb_2S_3 , XRD analysis did not have the signal of Sb_2S_3 . As explained in Q8, Sb_2S_3 remained amorphous in MSbNSs-3 and -4 as the

reaction temperature at 170 °C (the blue line is Sb_2S_3).

10. The claim on the four steps was also insufficient. For example, the authors said that “Step (2) and Step (3) were further confirmed by measuring the Fourier- transform infrared spectroscopy (FTIR) spectra of the reaction solutions (Fig. 2d). O-H stretching of alcohol (3550-3200 cm^{-1}) and S=O stretching of SO_2 (1350-1300 cm^{-1}) were tested to be stronger as the increase of O_2 amount, especially in the solvent of MSbNSs-4.” In fact, DDT might also be oxidized by O_2 to generate O-H and S=O. To address this question, reactions without adding DDT should be performed.

A: Thank you for your kind suggestion. It is worth noting that DDT here was acted as a cosolvent of SbCl_3 , so the method did not work in the absence of DDT. To deepen the understanding of the impact of DDT, the concentration of DDT was varied while the other experimental parameters were kept constant. As shown in Fig. S7, when decreasing the volume of DDT to 0.3 mL, the pore size and surface area of MSbNSs decreased. At the same time, the monodispersity of the products became worse possibly due to the decreased solubility of SbCl_3 . As the concentration of DDT increased, the obtained MSbNSs showed obvious mesoporous structures and good monodispersity. When the volume of DDT reaching to 2.0 mL, the diameter of the obtained MSbNSs decreased significantly, and numerous small nanoparticles appeared, which is ascribed to the excessive etching of mesoporous structures. All the results demonstrated that the DDT played a remarkable role in the process of the formation of the mesopores, which is consistent with our proposed formation mechanism of MSbNSs. We also added the necessary discussion in the manuscript.

11. Why the author chose 1210 nm to conduct photothermal therapy and laser-induced degradation, are other wavelengths of laser applicable?

A: Thank you for your kind concern. The wavelengths of 1064 nm/ 1210 nm/ 1470 nm are commonly used laser excitation wavelength. Considering that the maximum absorption of MSbNSs is around 1210 nm, choosing 1210 nm as excitation wavelength is helpful to get high

photothermal conversion of MSbNSs. This is also one of the novelties of our manuscript by applying a new wavelength for PTT and get excellent photothermal performance. Other wavelengths, such as the typical 1064 nm, are applicable as shown in Fig. S12.

12. In Fig. 3a and Fig. S7 and S8, the temperature values upon laser irradiation should be presented.

A: Thank you for your kind suggestion. We have provided the temperature values in Fig. 3a and Fig. S10 and S11.

13. “Hot channel” is another proposed mechanism to distinguish the differences of MSbNSs-2 and -3. Could temperature values be determined in Fig. 3c,d to support the “stronger localization of heat generation” between different MSbNSs? More details would be better to explain the differences.

A: Thank you for your kind suggestion. The simulation results can only qualitatively compare the heat power densities generated in different positions but cannot determine the temperature values. Moreover, the model we applied in the simulation cannot perfectly match the real mesoporous structure in MSbNSs. We have provided the simulation details in SI and added more discussions to explain the differences of “hot channel” in MSbNSs-2 and -3.

14. Why the PEGylation in Fig. 4a had such a big influence on the Vis-NIR absorption of MSbNSs with the strongest absorption peak moved to be 1480 nm?

A: Thank you for your concern. The red shift of the strongest absorption peak was ascribed to the small amount of aggregation of nanoparticles during the transition from oil phase to water phase by PEG modification.

15. Considering the liability of MSbNSs in vivo, the time-dependent biodistribution and long-term toxicity should be studied.

A: Thank you for your kind suggestion. We have provided the biodistribution and long-term toxicity evaluation as shown in Fig. S15, 16, 20.

16. The authors should mark all the figures with bars in Fig. 5a,j and Fig. S10 and S13.

A: Thank you for your kind suggestion. We have marked all the figures with bars in Fig. 5a,j and Fig. S10 and S13.

Reviewer #2 (Remarks to the Author):

In this manuscript, a novel selective etching method to fabricate semimetallic mesoporous nanostructures was successfully established which could be used as efficient multimodal nanoplatforms for theranostics. Overall, this is a well-organized work, which can be considered for publication after the authors address the following issues.

A: We appreciate Reviewer #2 for the confirmation of the novelty and significance of our manuscript.

1. It would be beneficial to further demonstrate the in vivo biodistribution of the MSbNSs.

A: Thank you for your kind suggestion. We have provided the in vivo biodistribution of MSbNSs in Fig. S15, 16.

2. The stability of MSbNSs in vivo should also be measured.

A: Thank you for your kind suggestion. We have evaluated the in vivo stability of MSbNSs in Fig. S17, 18. There was no significant PA signal change without 1210 nm laser irradiation, indicating a relevantly good in vivo stability of MSbNSs.

3. Some related references may be helpful to improve the introduction section of this manuscript.

A: Thank you for your kind suggestion. We have provided more references and discussions in the introduction.

Reviewers' Comments:

Reviewer #1:

Remarks to the Author:

Qu and co-authors have provided sufficient data and results to address our previous concerns, which were solid and convincing. Overall, their work reported a good strategy for preparing monodispersed mesoporous antimony nanospheres through a selective oxidation method with excellent theranostic performance. Thus, we have no more questions and recommend this manuscript to be published in Nature Communications without further revisions.

Reviewer #2:

Remarks to the Author:

The authors have fully addressed my concerns. I suggest the acceptance of this work.

Reviewer #3:

Remarks to the Author:

The authors presented a strategy for preparing monodispersed nanospheres (MSbNSs), using a partial oxidation of Sb nuclei and selective etching of the as-formed Sb₂O₃. By changing the experimental conditions, the authors obtained different nanostructures with near-infrared absorption properties for near-infrared laser-based cancer theranostics. Furthermore, DOX was loaded in the pores for cancer therapy after laser-induced biodegradation of MSbNSs. The work is very interesting. The fabrication and characterization of the MSbNSs are well presented and discussed. The in vitro and in vivo studies are less convincing and has major flaws. As such, the current version of the paper is not suitable for publication in NC.

- Authors have to demonstrate the loading degree of DOX inside the MSbNSs. Also, the release in serum, and at pH = 5.0/6.0 should be provided since this is typically the pH in the cancer cells.
- Authors have to demonstrate DOX is stable inside the pores and the distribution of the drug inside the pores of the MSbNSs.
- Authors have to demonstrate the drug is mainly inside the pores and not crystallizing on the surface.
- Authors have to demonstrate the degradation mechanism by providing evidence on the degradation of the MSbNSs, e.g., by TEM or other.
- Authors have to demonstrate the drug release is specific at the wavelength claimed in the work, by providing evidence of other laser irradiation intensities and the drug release of those results.
- Authors claim the internalization was mainly by endocytosis, but Fig. 5a is not enough to claim this. Please, provide additional data supporting this claim.
- In Fig. 5b, if the concentration in the graphic is of the MSbNSs, how can the authors be sure the same amount of drug is present in all samples? Provide evidence the loading degrees for DOX for the tested samples. Authors have also to compare the effect with pure DOX (control group with drug alone).
- Authors have to discuss the mechanism for the accumulation of the MSbNSs in the tumors. In Figs. 5f and S16, the quantification for the tumor accumulation must be provided. As the authors stated, most of the PEGylated MSbNSs were mostly accumulated in liver, spleen and kidney. Thus, this nanosystem does not represent an advancement for many other porous structures, such as silica and silicon based, regarding the in vivo drug delivery.
- Fig. 5 is just difficult to follow. All the time points tested are different, and thus, any conclusions not possible. Fig 5c and Fig. 5d should have similar time points for comparison. In Fig. 5f, the tumor accumulation quantification is missing.
- Looking to Fig. 5g, clearly the effect is not from the drug: as the authors demonstrate samples MSbNSs+L and MSbNSs/DOX+L achieved similar results.
- Authors should provide evidence that after 20 days there is no recurrence of the tumors.
- The discussion is very poor, and should provide more evidence and comparison with other porous based materials, and be more precise on the in vivo outcomes based solely on the results shown in

the paper.

- Authors should also analyze the immunologic aspects of the MSbNSs in vivo.

Reply to Reviewers

Manuscript ID: NCOMMS-21-18564A-Z

Manuscript Title: Degradable mesoporous semimetal antimony nanospheres for near-infrared
II multimodal theranostics

We are glad to learn that Reviewers #1, #2 & #3 did appreciate the novelty and significance of our manuscript. The comments posed by Reviewer #3 are helpful to improve our manuscript. We are also thankful for offering us a valuable revision opportunity to address the questions and comments posed by Reviewer #3. Amendments have been made as per the request of Reviewer #3, and we hope that the amendments will meet with your final approval for publication in Nature Communications. All the amendments are marked in a yellow background in the revised manuscript and Supplementary Information. Our replies to the reviewers' comments are listed below point-by-point.

Reviewer #1 (Remarks to the Author):

Qu and co-authors have provided sufficient data and results to address our previous concerns, which were solid and convincible. Overall, their work reported a good strategy for preparing monodispersed mesoporous antimony nanospheres through a selective oxidation method with excellent theranostic performance. Thus, we have no more questions and recommend this manuscript to be published in Nature Communications without further revisions.

A: We appreciate Reviewer #1 for confirming the general interest and significance of our manuscript and recommending the acceptance of our manuscript.

Reviewer #2 (Remarks to the Author):

The authors have fully addressed my concerns. I suggest the acceptance of this work.

A: We appreciate Reviewer #2 for confirming the general interest and significance of our

manuscript and recommending the acceptance of our manuscript.

Reviewer #3 (Remarks to the Author):

The authors presented a strategy for preparing monodispersed nanospheres (MSbNSs), using a partial oxidation of Sb nuclei and selective etching of the as-formed Sb_2O_3 . By changing the experimental conditions, the authors obtained different nanostructures with near-infrared absorption properties for near-infrared laser-based cancer theranostics. Furthermore, DOX was loaded in the pores for cancer therapy after laser-induced biodegradation of MSbNSs.

The work is very interesting. The fabrication and characterization of the MSbNSs are well presented and discussed. The in vitro and in vivo studies are less convincing and has major flaws. As such, the current version of the paper is not suitable for publication in NC.

A: We appreciate Reviewer #3 for confirming the general interest and significance of our manuscript.

- Authors have to demonstrate the loading degree of DOX inside the MSbNSs. Also, the release in serum, and at pH = 5.0/6.0 should be provided since this is typically the pH in the cancer cells.

A: The loading efficiency of DOX into MSbNSs was shown in Fig. 2f. We have also added the release profile in PBS (pH=5.0/6.5) as shown in Supplementary Fig. 13

- Authors have to demonstrate DOX is stable inside the pores and the distribution of the drug inside the pores of the MSbNSs.

A: We have measured the elemental mapping of Cl element in DOX to demonstrate the distribution of DOX inside the pores of DOX-loaded PEGylated MSbNSs-3 as shown in Supplementary Fig.12.

- Authors have to demonstrate the drug is mainly inside the pores and not crystallizing on the surface.

A: We have added the elemental mapping of Cl element in DOX-loaded PEGylated MSbNSs-3 to show the distribution of DOX as shown in Supplementary Fig. 12.

- Authors have to demonstrate the degradation mechanism by providing evidence on the degradation of the MSbNSs, e.g., by TEM or other.

A: The TEM images to show the degradation of MSbNSs were shown in Fig. 3b.

- Authors have to demonstrate the drug release is specific at the wavelength claimed in the work, by providing evidence of other laser irradiation intensities and the drug release of those results.

A: Thank you for your kind suggestion. It is noteworthy that the on-demand release of MSbNSs can be observed under laser irradiation with other wavelengths long as it can produce photothermal effect. We have added the drug release profile under the laser irradiation of 1064 nm wavelength as shown in Supplementary Fig.14.

- Authors claim the internalization was mainly by endocytosis, but Fig. 5a is not enough to claim this. Please, provide additional data supporting this claim.

A: The phenomenon that nanocarriers uptaken by cells through endocytosis has been widely reported. The confocal microscopy was often used to characterize the cellular internalization. As shown in Figure 5a, the fluorescence of DOX loaded in MSbNSs was co-located with the fluorescence of Lyso-Tracker in panc02 cells, indicating endocytosis of the PEGylated MSbNSs. To further confirm the endocytosis process, we have added flow cytometry analysis to support this claim as shown in Supplementary Fig. 17.

- In Fig. 5b, if the concentration in the graphic is of the MSbNSs, how can the authors be sure the same amount of drug is present in all samples? Provide evidence the loading degrees for DOX for the tested samples. Authors have also to compare the effect with pure DOX (control group with

drug alone).

A: In Fig. 5b, we used the same sample for group MSbNSs/DOX and MSbNSs+DOX+L, the only difference is with/without laser irradiation. Considering the different amounts of DOX and different cellular uptake efficiencies of pure DOX and MSbNSs/DOX, we think the comparison of cell viability between pure DOX and MSbNSs/DOX is not considered scientifically rational.

- Authors have to discuss the mechanism for the accumulation of the MSbNSs in the tumors. In Figs. 5f and S16, the quantification for the tumor accumulation must be provided. As the authors stated, most of the PEGylated MSbNSs were mostly accumulated in liver, spleen and kidney. Thus, this nanosystem does not represent an advancement for many other porous structures, such as silica and silicon based, regarding the in vivo drug delivery.

A: Thank you for your concern. We have provided the accumulation rate of PEGylated MSbNSs in the tumors as shown in Supplementary Fig.20. The key advancement of our manuscript is to develop a new selective etching method to fabricate semimetallic mesoporous nanostructures and their potential in multimodal theranostics. Indeed, in term of in vivo drug delivery, silica-based nanosystems have excellent drug loading abilities due to their ordered pore structure and large surface area.

- Fig. 5 is just difficult to follow. All the time points tested are different, and thus, any conclusions not possible. Fig 5c and Fig. 5d should have similar time points for comparison. In Fig. 5f, the tumor accumulation quantification is missing.

A: Fig. 5c and Fig. 5d are different figures to demonstrate different properties. Fig. 5c is the in vivo photoacoustic imaging change to show the accumulation condition of PEGylated MSbNSs, which needs longer time to 12 h and is used to decide the optimal irradiation time. Fig. 5d is showing the temperature monitoring at the tumor sites by the irradiation of 1210 nm laser, which should not exceed 10 min (7 min) in our case. These two figures are not comparable. Fig. 5f showed the biodistribution of PEGylated MSbNSs after PTT, while the tumor

accumulation quantification of PEGylated MSbNSs without irradiation was added in Supporting information Fig.20.

- Looking to Fig. 5g, clearly the effect is not from the drug: as the authors demonstrate samples MSbNSs+L and MSbNSs/DOX+L achieved similar results.

A: Thanks for your concern. Fig. 5g demonstrated that MSbNSs+L can eliminate the tumors, which is the main effect. However, MSbNSs/DOX+L group will fasten the elimination of tumors and show synergistic effect, indicated by the smaller average tumor volume compared to that in MSbNSs+L group.

- Authors should provide evidence that after 20 days there is no recurrence of the tumors.

A: Thanks for your concern. We did not observe any recurrence of tumors after 20 days as shown in the survival rate test.

- The discussion is very poor, and should provide more evidence and comparison with other porous based materials, and be more precise on the in vivo outcomes based solely on the results shown in the paper.

A: Thanks for your kind suggestion. We have added more discussion about the comparison with other mesoporous based materials, including silicon or carbon-based materials, in the manuscript.

- Authors should also analyze the immunologic aspects of the MSbNSs in vivo.

A: PTT has been reported to elicit immunogenic cell death by inducing dying tumor cells to release damage-associated molecular patterns, which can lead to an adaptive antitumor immune response. This phenomenon has been widely reported. Furthermore, our previous work (Adv. Mater. 2021, 33, 2100039) about photothermal/immunotherapy using Sb

nanopolyhedrons has also demonstrated that PTT will generate mild immune response as widely reported. The key advancement of our manuscript is to develop a new selective etching method to fabricate semimetallic mesoporous nanostructures and their potential in multimodal theranostics. The immunologic aspects of the MSbNSs in vivo will be shown in our future work.

Reviewers' Comments:

Reviewer #3:

Remarks to the Author:

The current work by Chen et al. describes the synthesis of Sb based mesoporous NPs for anticancer drug delivery. Despite there are some interesting findings in the materials fabrication part, the publication of the current work to Nature Communication is too premature due to the major flaws regarding to the experimental design of the work, and some of the current experimental results cannot fully support their conclusions.

1. The NPs fabrication and synthesis procedure is relatively new and the each step is well characterized; however, this is definitely not "...the first report of semimetallic mesoporous structure...". A little thoroughly checking in the literature shows that simply for silicon NPs, both bottom-up and top-down production of mesoporous silicon NPs with altered porosity and surface area have long been established. In addition, the concept of precursor---nuclei formation---intermediate product---etching---nuclei collapsing to form porous structure is also not new (Bottom-up synthesis of high surface area mesoporous crystalline silicon and evaluation of its hydrogen evolution performance. Nature communications, 2014, 5, 3605; An iron silicate based pH-sensitive drug delivery system utilizing coordination bonding" J. Mater. Chem. B 2013, 1, 2837-2842).
2. The current system is not necessarily suitable for anti-cancer applications. For pharmaceutical engineering or biological engineering, something new does not mean something good or something suitable. Despite the fact that according to U.S. Centers for Disease Control and Prevention and U.S. Environment Protection Agency, antimony compounds are allocated as hazard substances, and the the Reference Dose (RfD) for antimony is only 0.0004 mg/kg/d, which is hundreds times lower than the dosage applied in the current study. Simply as a drug carrier, it did not show huge promote or advantages comparing to previous studies in regarding to drug loading efficiency, controlled release manner or photothermal conversion efficiency, etc. Therefore, the current system neither resolve any practical obstacles or challenges, nor providing new concepts for further cancer treatment. The novelty of the materials is not connected to the application scenario.
3. Authors have to further explain why the absorption peak will be red-shifted from MSbNSs-1 to MSbNSs-3, whereas MSbNSs-4 lost the NIR-II absorption.
4. The observed pH-dependent DOX release may be simply due to the protonation/deprotonation of DOX.
5. The plasma stability and integrity of Sb NPs under 37°C should be evaluated.
6. TUNEL is mainly for evaluating apoptosis but not necrosis. Fig. 5j does not look like the true and actual staining of TUNEL but rather due to the lack of property staining and imaging process.
7. For in vivo study, plain NPs without light irradiation should be added.
8. From the in vivo study, the major toxicity of the NPs comes from the photothermal efficiency of plain NPs, then what is the advantages of adding extra anti-cancer drugs.
9. The authors applied panc02 (pancreatic carcinoma cells) as model cells, whereas for human, pancreas is deeply buried in retroperitoneal space and surrounded by intestine, for murine, the pancreas is fully covered by spleen. Thus, whether NIR-II light can effectively penetrate into the proper cite is the key challenge to achieve the desired treatment efficacy, whereas the authors did not provide the corresponding information and detailed explanation for all these issues.
10. Since the major cellular toxicity is from the physical burning, therefore a more detailed sub-organ distribution of the NPs is crucial. Flow cytometry should be used to quantify the percentage of NPs interacting cells within defined cell type populations (cancer cells, B cells, T-reg cells, T-killer cells, macrophages, dendritic cells, etc.) of tumors, and orthotopic xenograft is preferred and should be added to the study to confirm the system has a valid biological application in vivo.

Reviewer #3

The current work by Chen et al. describes the synthesis of Sb based mesoporous NPs for anticancer drug delivery. Despite there are some interesting findings in the materials fabrication part, the publication of the current work to Nature Communication is too premature due to the major flaws regarding to the experimental design of the work, and some of the current experimental results cannot fully support their conclusions.

1. The NPs fabrication and synthesis procedure is relatively new and the each step is well characterized; however, this is definitely not "...the first report of semimetallic mesoporous structure...". A little thoroughly checking in the literature shows that simply for silicon NPs, both bottom-up and top-down production of mesoporous silicon NPs with altered porosity and surface area have long been established.

In addition, the concept of precursor---nuclei formation---intermediate product---etching---nuclei collapsing to form porous structure is also not new (Bottom-up synthesis of high surface area mesoporous crystalline silicon and evaluation of its hydrogen evolution performance. Nature communications, 2014, 5, 3605; An iron silicate based pH-sensitive drug delivery system utilizing coordination bonding" J. Mater. Chem. B 2013, 1, 2837-2842).

A: The examples Reviewer #3 has mentioned are actually based on silicon or iron, not on semimetallic elements, which is definitely not contradictory to our claim that *this is the first report of **semimetallic** Sb mesoporous structure.*

2. The current system is not necessarily suitable for anti-cancer applications. For pharmaceutical engineering or biological engineering, something new does not mean something good or something suitable. Despite the fact that according to U.S. Centers for Disease Control and Prevention and U.S. Environment Protection Agency, antimony compounds are allocated as hazard substances, and the the Reference Dose (RfD) for antimony is only 0.0004 mg/kg/d, which is hundreds times lower than the dosage applied in the current study. Simply as a drug carrier, it did not show huge promote or advantages comparing to previous studies in regarding to drug loading efficiency, controlled release manner or photothermal conversion efficiency, etc. Therefore, the current system neither resolve any practical obstacles or challenges, nor providing new concepts for further cancer treatment. The novelty of the materials is not connected to the application scenario.

A: Our MSbNSs system has been demonstrated to possess multifunctionality and excellent NIR-II photothermal performance with a relatively new excitation wavelength of 1210 nm and a high NIR-II PCE of ~44%. These properties do provide better therapeutic outcomes and emphasize the novel properties of our material system. Based on the unique semimetallic mesoporous structures, more applications requiring mesoporous structure and large surface area in energy storage, catalysis etc. can be further explored and demonstrated.

3. Authors have to further explain why the absorption peak will be red-shifted from MSbNSs-1 to MSbNSs-3, whereas MSbNSs-4 lost the NIR-II absorption.

A: Sb has the property of localized surface plasmon resonances (LSPR), which will change the absorption properties with the morphology change. From MSbNSs-1 to MSbNSs-3, the pore sizes will be larger and this will lead to a red-shifted NIR-II absorption. For MSbNSs-4, the majority of the component has been oxidized to Sb_2O_3 , which does not have NIR-II absorption.

4. The observed pH-dependent DOX release may be simply due to the

protonation/deprotonation of DOX.

A: We don't deny that the protonation/deprotonation of DOX might be one of reasons. Indeed, pH-dependent DOX release has been widely reported in different mesoporous systems. The stimuli-responsive drug release derived from NIR laser irradiation induced structural collapse is the focus of this work, which has been totally studied as shown in Fig. 3g.

5. The plasma stability and integrity of Sb NPs under 37°C should be evaluated.

A: Please see the results as shown in Fig. S10. SbNSs-1 still showed good stability even the temperature of the solution reached ~80°C. Furthermore, we have carefully studied the relationship between the porosity of MSbNSs and their stability as shown in Fig. 3, Fig. S9 and Fig. S10.

6. TUNEL is mainly for evaluating apoptosis but not necrosis. Fig. 5j does not look like the true and actual staining of TUNEL but rather due to the lack of property staining and imaging process.

A: The TUNEL data in Fig.5j have been updated.

7. For in vivo study, plain NPs without light irradiation should be added.

A: MSbNSs without light irradiation data in vitro has proven that MSbNSs have high cell viability and the MSbNSs/DOX without laser irradiation was performed as the control group in vivo. Please see Fig. 5b.

8. From the in vivo study, the major toxicity of the NPs comes from the photothermal efficiency of plain NPs, then what is the advantages of adding extra anti-cancer drugs.

A: Loading of extra anti-cancer drugs will accelerate the elimination process as

explained in the manuscript. Loading of different types of drugs will have different multimodal synergistic phototherapies, such as chemotherapy, photodynamic therapy and photo-immunotherapy with PTT to avoid heat shock of cancer cells and eliminate the possible residual cancer cells.

9. The authors applied panc02 (pancreatic carcinoma cells) as model cells, whereas for human, pancreas is deeply buried in retroperitoneal space and surrounded by intestine, for murine, the pancreas is fully covered by spleen. Thus, whether NIR-II light can effectively penetrate into the proper cite is the key challenge to achieve the desired treatment efficacy, whereas the authors did not provide the corresponding information and detailed explanation for all these issues.

A: We just chose panc02 cells as a demonstration of cancer cells applied in this study. One of the main advantages of NIR-II excitation is deep tissue penetration, which has been proven and widely accepted by many reports (Wu et al., *Nano-Micro Letters* 2020, 12, Chitgupi et al., *Adv. Mater.* 2019, 31, 1902279).

10. Since the major cellular toxicity is from the physical burning, therefore a more detailed sub-organ distribution of the NPs is crucial. Flow cytometry should be used to quantify the percentage of NPs interacting cells within defined cell type populations (cancer cells, B cells, T-reg cells, T-killer cells, macrophages, dendritic cells, etc.) of tumors, and orthotopic xenograft is preferred and should be added to the study to confirm the system has a valid biological application in vivo.

A: The distribution of MSbNSs in the organs has been added in the last revision requested by Reviewer#3. Sub-organ distribution of MSbNSs will be studied in our future work. Our present manuscript is not involved with immunotherapy, but it is not ruled out that immunotherapy will be studied in our future work. Anyway, we still thanks for your kind suggestions.